# Factors Determining Plasticity of Responses to Drugs

**DOI:** 10.3390/ijms23042068

**Published:** 2022-02-13

**Authors:** Michael J. Parnham, Jennifer A. Kricker

**Affiliations:** 1EpiEndo Pharmaceuticals ehf, Bjargargata 1, 102 Reykjavik, Iceland; jk@epiendo.com; 2Pharmacology Consultant, 65812 Bad Soden am Taunus, Germany

**Keywords:** pharmacological plasticity, drug discovery, receptor expression, metabolic enzymes, cell plasticity, tolerance, chronopharmacology, duration of action, confounding factors, biomarkers

## Abstract

The plasticity of responses to drugs is an ever-present confounding factor for all aspects of pharmacology, influencing drug discovery and development, clinical use and the expectations of the patient. As an introduction to this Special Issue of the journal *IJMS* on pharmacological plasticity, we address the various levels at which plasticity appears and how such variability can be controlled, describing the ways in which drug responses can be affected with examples. The various levels include the molecular structures of drugs and their receptors, expression of genes for drug receptors and enzymes involved in metabolism, plasticity of cells targeted by drugs, tissues and clinical variables affected by whole body processes, changes in geography and the environment, and the influence of time and duration of changes. The article provides a rarely considered bird’s eye view of the problem and is intended to emphasize the need for increased awareness of pharmacological plasticity and to encourage further debate.

## 1. Introduction

Whether as a patient taking a prescribed medicine, a laboratory researcher assessing the results of an experiment or a physician considering the best therapy to prescribe for a patient, most of us are aware that pharmaceutical therapies do not always give the responses we want or expect. The challenge lies in the fact that the search for new drugs, the development of robust test conditions and the assessment of clinical conditions that may affect responses to drugs are all subject to many confounding factors. These interfering variables can markedly influence the desired or expected drug response leading to pharmacological plasticity. This term is defined very broadly in this review as the outcome of the influence of any internal or external variables or processes on the presence or extent of the desired response to a therapeutic drug. Such influences can arise from poorly controlled conditions, genetic background, ongoing disease processes or variability within the studied population, whether cells, animals or human subjects. As scientists, we recognize that variability can be controlled when the mechanisms are understood, investigational approaches standardized, and internationally accepted guidelines introduced and followed. Consequently, many proposals have been made in recent years to improve the reproducibility and quality of drug testing, to enhance translatability to the clinic and to increase understanding of the factors that modify clinical responses to drugs [1,2,3,4,5,6]. However, science is essentially reductionistic and we all try to simplify our approaches to scientific issues. It is, therefore, crucial that awareness is raised about the various factors that contribute to the complexity of pharmacological plasticity and the approaches that can be taken to control them, so that the interpretation of pharmacological responses is facilitated. A first attempt to explain the breadth of plasticity was made in a recent article [7] and the publication of this Special Issue on pharmacological plasticity draws attention to some of these matters. The present reviewaddresses some of the multiple levels of pharmacological plasticity and clinical variables to consider, as an introduction to the subject, together with approaches taken to limit plasticity. The discussion is by no means comprehensive but offers a bird’s eye view and is intended to encourage further debate.

## 2. Levels of Plasticity

As shown in Table 1, there are several different levels of potential plasticity in responses to drugs.

### 2.1. Molecular Structure

The first level is perhaps the most obvious, namely, variations in the structure of drugs and their receptors or target molecules. For more than a century, chemists and more recently, molecular biologists and protein engineers have been systematically modifying drug structures and analyzing structure–activity relationships (SAR) to improve and optimize drug responses or reduce metabolism [8,9]. The use of computer-assisted drug design has vastly enhanced this process, which is still driven by the chemist’s own assessment of the likelihood of a pharmacophore being therapeutically effective. Such selection has become possible, independent of structural analysis of ligand docking to the receptor, for instance, using deep learning networks based on large-scale data, to enable drug–target interaction prediction and de novo drug design, as described by Wankyu Kim in this *IJMS* Special Issue [10]. Nevertheless, outcomes are still not entirely predictable. This is seen from the recent finding that the Curevac COVID-19 vaccine, which used unmodified mRNA, was not as effective as expected, in comparison to the other two widely used mRNA vaccines, which incorporated an mRNA nucleotide called pseudouridine instead of uridine to reduce inflammatory side effects [11]. Because of side effects, the Curevac vaccine had to be used at a lower dose than planned. To avoid such costly late-stage disappointments in the development of a pharmaceutical, it is recommended to introduce rigorous robust methodology in the initial preclinical compound screening and target validation [2].

Another widely taken approach is to develop allosteric modulators, which modulate a molecular target protein by acting at a site remote to the endogenous ligand binding site. Such drugs include the anxiolytic/sedative benzodiazepines which act allosterically at ion-linked GABA-A receptors [12], and some compounds acting at nicotinic acetylcholine receptors which alter receptor folding [13]. These agents often have the advantage that they avoid adverse events related to direct target binding, as found with cannabinoid antagonists [14]. Such off-target binding of a drug can also induce undesired adverse effects, but unexpected actions can also lead to potential novel uses, as indicated by Li et al. in an article in this *IJMS* Special Issue [15].

### 2.2. Expression of Drug Target Genes

A major source of variation in drug responses is the individual alteration in the expression of therapy relevant target genes [16]. This can range from mutations in infectious agents, leading to altered virulence—as with the SARS-CoV-2 virus by which we have all been affected—to changes in disease-related target enzymes. This whole field has recently been reviewed for responses to anticancer drugs [17].

On the one hand, this altered gene expression can be due to somatic mutations, resulting in polymorphic differences between individuals in the structure and expression of drug target proteins, such that the same drug acting at these receptors will cause more pronounced or less pronounced effects or no effect at all if the protein is absent or fundamentally altered. For instance, there is considerable genetic polymorphism in dopamine receptors, leading to differences in Parkinson’s disease susceptibility and to varying responses to drug therapy [18]. Similarly, polymorphisms in several different genes have been shown to determine responses to interferon-beta in multiple sclerosis [19]. As discussed below, this heterogeneity can also arise between ethnic groups or as a result of disease. A further level of genetic variability in drug targets can arise as a result of epigenetic modification of gene transcription and translation, arising from DNA methylation or histone acetylation, for example. This can be the result of lifestyle, disease, aging or even the use of medications. A review of the epigenetic mechanisms and implications of drugs, both in terms of epigenetic changes which can affect drug targets and the capacity of drugs to induce epigenetic changes, is presented by Kringel et al. in this *IJMS* Special Issue [20].

### 2.3. Drug Metabolism

The various enzymatic and non-enzymatic pathways by which drugs can be broken down in the body are, to a large extent, well-defined [21]. These can be circumvented partially by using different routes of administration or by pharmaceutical formulation, for instance as delayed release or slow-release products, nanoparticles or by targeting to specific anatomical sites [22,23].

However, metabolic pathways are subject to a wide range of factors that promote variability or plasticity, also leading to drug–drug interactions through mutual interference with the metabolism of the other drug. Interfering factors include individual variations with under- or over-expression of genes for metabolic enzymes and polymorphic forms of the enzymes, particularly of cytochromes CYP2D6, CYP2C9, CYP2C19 and others in humans. This has led to the introduction of methods to genotype patients for CYP polymorphisms before drug therapy, but the large individual variability, especially in CYP2D6, restricts precision and has limited the use of such tests [24].

Considerable differences also exist between males and females in factors affecting drug responses [7,25]. This is clearly apparent in relation to metabolism, as hepatic metabolizing enzymes can show pronounced sex differences, particularly CYP3A4 in women [26].

Other sources of variability in metabolism of drugs include feedback reactions of metabolic products on the metabolic pathway and the generation of active, toxic or reactive metabolites. These have generally been approached using analysis of chemical structures by liquid chromatography-mass spectrometry, but this has often failed to establish a clear link between reactive metabolite and tissue injury which requires integration of knowledge on chemical structure, safe and toxic pathways [27]. In addition, a variety of external influences can modify drug metabolism, such as diet or exercise, as discussed elsewhere in this *IJMS* Special Issue [28]. Often, particularly in the elderly, the issue is confused further by exposure to polypharmacy leading to metabolic drug interactions, in association with age-related changes [29].

For many patients, it is difficult to understand why the medications they are taking do not appear to work or exert unexpectedly pronounced responses and such individual metabolic plasticity is one of the major reasons for this. An excellent detailed review of factors affecting and confounding the interpretation of drug analyses in various tissues of the body, including patient non-compliance with the therapy, has been published recently [30]. The increasing use of tissue or liquid biopsy samples permits the determination of drug metabolizing enzyme inhibition ex vivo, the study of genotype-phenotype associations, the evaluation of tissue expression profiling following an inducer, and the assessment of correlations between tissue expression profiles and in vivo-derived trait measures, such as biomarker plasma levels and drug pharmacokinetics (PK) [31]. Such data are invaluable for the prediction of potential drug–drug interactions, toxicity or altered efficacy.

### 2.4. Cell Plasticity

The next level of plasticity occurs at the cellular level, with phenotype changes, malignant transformation and the influence of disease processes and cellular senescence. Many cell types in the body can adjust to changes in the tissue milieu. This is particularly true of blood cells and those mediating innate and adaptive immune responses. Circulating monocytes, for instance can differentiate to tissue macrophages, to microglial cells in nervous tissue and Kupffer cells in liver on leaving the blood stream (and the phenotype may be locally self-sustaining), with altered properties to promote innate defense or maintain nerve function [32].

However, the same cells can undergo a phenotype change, depending on locally generated cytokines, to generate more inhibitory M2 macrophages or around tumors to form relatively quiescent tumor-associated macrophages, which respond differently to immunomodulating drugs. In fact, a number of drugs, particularly when targeted to macrophages, have been shown to modify these phenotype changes in macrophages, either facilitating resolution of inflammation or alleviating the tumor-associated immunosuppressive environment [33,34]. This is the case, for instance, with macrolide antibiotics such as azithromycin. In healthy individuals, azithromycin initially stimulates peripheral blood neutrophil activity, promoting host defence, and later dampens neutrophil activity. In macrophages activated during inflammation, azithromycin is able to change the inflammatory macrophage M1 phenotype to an anti-inflammatory M2 phenotype and promote resolution of inflammation [35]. Equally, the γδ-T lymphocytes, which have both innate and adaptive immune functions, alter their properties in varying milieux [36]. Interestingly, when injured, as in SARS-CoV-2 infection, endothelial cells lining the blood vessel lumen can also undergo a change to a procoagulant phenotype, generating inflammatory mediators and responding to coagulant factors which promote thrombosis and thus, become more susceptible to drug therapy [37]. Similar changes in cellular sensitivity are also well known in spinal primary afferent neurones, in which inflammation or mechanical injury cause increased excitation (wind-up), with enhanced sensitivity to painful stimuli, which can become chronic [38]. This wind-up can also be targeted by existing or potential analgesic drugs [39].

A crucial process in the plasticity of tissue cells is the epithelial-mesenchymal transition (EMT) whereby epithelial cells, normally organized in sheet-like arrangements of polarized cells connected with tight and adherent junctions, transition into mesenchymal cells with increased migratory capacity and plastic properties. These mesenchymal cells are characterized by invasiveness, mobility, and the ability to differentiate into other cell types, driving tissue growth and regeneration as well as fibrotic processes in some tissues [40]. Not only is this EMT process important for normal growth and development or the promotion of wound repair and tissue regeneration, it can also facilitate tumor progression and metastasis. As it is a hallmark of many cancers and organ fibrosis, many drug therapies aim to disrupt this plasticity by affecting either the inducers or effectors of EMT, or even target the reverse process mesenchymal-epithelial transition (MET) [41]. Despite extensive research, no drug has yet been identified which prevents liver or lung fibrosis. However, recently, it has been shown that changes in epithelial cells can be modified by drugs to promote epithelial barrier protection [42]. Fifteen-membered macrolides including azithromycin have been shown to suppress EMT [43] and have positive effects on TGFβ-induced fibrosis [44,45]. Ultimately, with increasing age and the presence of various stimuli such as telomere erosion, oncogene activation, excessive oxidative stress, and irradiation, cellular senescence occurs. This involves the activation of the DNA damage response (DDR) pathway which helps to safeguard against the accumulation of DNA damage, often recognized as the underlying mechanism of a wide variety of age-related pathologies including cancer [46]. Paradoxically, senescence can also be induced by cancer chemotherapeutic drugs, creating an immunosuppressive milieu by senescent tumor cells and facilitating tumor progression.

From this brief consideration of cell plasticity, it is clear that the processes of chronic disease progression, whether neuronal, infectious, inflammatory or malignant, are associated with marked changes in the type and characteristics of the cells involved, resulting in pronounced alterations in the sensitivity of these cells to drug therapy. As a consequence, responses to drugs can vary with age, with the stage of a disease and with the time of dosing. These aspects of time and disease-related drug susceptibility are discussed below. Essentially, this variability means that doses of the same drug may need to be altered with time or that different drugs will need to be used sequentially at different stages of the disease, as discussed in an earlier review [7].

Before moving on from the topic of cell plasticity, it should be pointed out that recent research has found that ribosomes involved in protein translation are not specific for a defined mRNA but are recruited to mRNAs from a homogenous intracellular pool and are specialized for particular cell and tissue types. Consequently, it is likely that ribosomes may be altered by disease processes and changes in cellular states. Apart from the fact that this will result in changes in drug target molecules and potential drug responses, it also opens up the possibility of developing drugs that target specific ribosomes associated with pathological processes [47].

### 2.5. Tissue Plasticity

It is difficult to clearly distinguish between cell and tissue plasticity because the processes are closely integrated. However, for the purpose of this article, it is helpful to consider separately a few internal processes which affect organ systems and the whole organism.

Cardiovascular changes, of course, occur as the blood provides crucial nutrients and oxygen, removing metabolic waste products, for every organ system. Consequently, metabolic diseases, such as diabetes are closely linked with cardiovascular changes and therapeutic drugs interact with both vascular and metabolic responses [48]. Immune responses and vascular responses are also closely integrated, not only via hyperoxic or ischemic conditions, but also in terms of leukocyte trafficking in various types and phases of inflammation [49]. This is seen in atherosclerosis, myocardial infarction and also COVID-19, which involve the immune system, endothelium and coagulation [50,51,52].

However, the nervous system is just as pervasive; even non-innervated cells, such as circulating leukocytes and macrophages, also possess receptors for neurotransmitters and are subject to neuronal modulation, for instance in cardiovascular disease [53]. Antipsychotic drugs have, in fact, been shown to affect inflammation by modifying inflammatory cytokine release [54]. Consequently, cardiovascular and psychoneuronal changes, including those in response to drugs, can have ramifications well beyond the physiological system which they target. Neuroendocrine responses are a good example, such as stress, depression and other mood changes, as well as central chronobiological and sex hormonal influences discussed below, which can have pronounced negative effects on immune defense, even leading to chronic inflammation [55]. Such interactions have been termed “immunoceptive interference” [56].

As a corollary, infections and immune responses can also affect neuronal function. Thus, it is becoming increasing apparent that viral infections during pregnancy can induce inflammation and epigenetic changes which can lead in the adult offspring to autism or schizophrenia [57]. These interactions have repercussions both for drug development and for the understanding of drug response plasticity. For instance, the balance between Wnt-β-catenin and Shh cell signaling pathways controls angiogenesis, neurodevelopment, central nervous system (CNS) morphogenesis and neuronal guidance, all of which are important to maintain a healthy blood–brain barrier. This signaling pathway has been proposed as a potential target for the treatment of autism spectrum disorder [58].

While we have previously touched on the effects of sex and diet on drug metabolism, at the level of the whole organism, it is important to point out that sex has become recognized as a crucial variable for all aspects of pharmacological or therapeutic responses, as well as physiology and pathology in animals and humans [25,59]. This has led to an emphasis on essential comparisons between males and females in all relevant pharmacological and therapeutic studies. The complex nature of the influence of sex is seen from the association between autoimmunity and sex. Autoimmunity occurs far more frequently in women than in men and is not just a question of hormones, X chromosome-linked immune response genes and other physiological factors, such as the inhibitory effect of pregnancy and the role of the placenta, play a role [60].

These whole-body interactions are important for the development and use of animal models. While many attempts are being made to reduce the use of laboratory animals with cell-cultures or co-cultures, organs-on-a-chip, organoids and other complex in vitro models, the systemic interactions discussed above can only be studied to a limited extent in such models [61,62,63]. On the other hand, other confounding factors can arise in whole animal models. Thus, while administration of the viral mimetic, poly(I:C) to pregnant mice has become widely used as a model for immune-mediated neurodevelopmental disorders and mental illnesses, differences in the batch of the poly(I:C) used can result in considerable differences in the immune response generated [64]. Because of these complex interactions, animal models, particularly those of sub-acute or chronic diseases, need very careful validation for interfering or confounding factors.

### 2.6. Environment

In addition to the different levels of predominantly internal physiological factors which affect the plasticity of drug responses, there are a variety of external environmental factors and interactions which impinge on the subject, particularly in a clinical context.

Crucial factors are geography and ethnicity. With regard to the latter, the genetic background of ethnic groups can vary markedly and in terms of drug responses, the differences between European, African and East Asian populations can be considerable. This is often related to the high polymorphism of the human leukocyte antigen (HLA) system, particularly for autoimmune disease susceptibility or hypersensitivity reactions to drugs [65]. For instance, with hypersensitivity to abacavir, a nucleoside analog reverse-transcriptase inhibitor of the HIV virus, association with HLA-B*57:01 has a 100% negative predictive value, such that a successful global pre-prescription screening strategy for this locus has been introduced [66].

While genetic polymorphisms account for some of the differences in responses to drugs between ethnic communities, other ethnic factors play a role, including diet or religious traditions. For instance, when Asian or African immigrants moved to Britain, their change in diet led to alterations in drug responses, particularly in drug half-lives in plasma [67]. Moreover, the ability of soy protein to inhibit induction of CYPs may not only contribute to the lower incidence of cancer and cardiovascular disease in East Asian countries, but will undoubtedly modify the PK of drugs in these populations [68].

In Western countries, there is such an intermingling of ethnicity, cultures, diets and traditions, particularly in cities, that such broad differences are less pronounced. However, international travel generates other issues (independently of the recent problems with SARS-CoV-2). In this regard, circadian rhythms are a crucial factor, not only in jetlag, but also in modifying drug responses, both therapeutic and toxic [69,70]. In addition to the effect of melatonin, released from the pineal gland under the influence of the light-sensitive suprachiasmatic nucleus in the brain, the presence of clock genes which oscillate in a circadian manner in most cells and tissues and regulate transcription of a large number of genes, accounts for considerable diurnal variation in PK and pharmacodynamic properties [71]. Thus, chronoefficacy of drugs varies for diverse indications including cancer, neuropathic pain, cardiovascular disease, and metabolic disease.

### 2.7. Time

Another general factor that was previously underemphasized in pharmacological investigations is time. In terms of age and aging, the ongoing growth and development process in children clearly has to be considered and this precludes the simple use of adjustment of drug dosing according to their lower body weight. In the elderly, it is well known that metabolic processes decline, cellular and tissue senescence kicks in and individuals become more sensitive to disease. However, in this respect, in both the young and the elderly, the duration and stage of the disease are important plastic factors to consider. As discussed above, cell phenotypes change as diseases progress, with the associated processes of tissue breakdown and fibrotic or cirrhotic changes. Most diseases can be self-limiting, if not progressing to chronic conditions, and active processes exist, such as inflammation resolution or exercise, which counteract the pathological progression. Consequently, drug treatment can be prophylactic, to prevent initiation of the disease, therapeutic to limit the disease, or remission or resolution inducing to delay or even prevent disease progression. Often, such temporal distinctions are not taken into account adequately and treatment tends to be based, at least initially, simply on dose level and the relief of symptoms.

However, in translating doses from preclinical to clinical studies, in which relationships between dose, plasma concentration and effect are crucial, hysteresis may occur in which there is a delay between concentration and action at the target and the subsequent response. This can result in skewing of the plasma concentration/drug effect relationship, due to a lag phase which limits access to the site of drug action, as with the vasodilator molsidomine, which is first converted to the NO-stimulating metabolite SIN-1, or to slow receptor kinetics, as with antihypertensive telmisartan, which dissociates slowly from binding to AT1 receptors [72].

Another issue with repeated or long-term treatment with many drugs, including analgesics, vasoconstricting decongestants, antidepressants and others is the occurrence of tachyphylaxis, a process whereby the effectivity of the drug decreases, which can usually be overcome by increasing the dose. With alpha-adrenergic decongestants, the tachyphylaxis is due to receptor internalization, but with opioids, the mechanism remains unclear [73]. With this class of analgesics, as well as with antidepressants, the tachyphylaxis can develop into tolerance, with prolonged loss of efficacy. Mitigating approaches are discussed in the next section.

Tolerance, with loss of response to drug treatment, is also particularly relevant for drugs affecting the immune system, as the immune system is highly adept at adjusting to altered conditions. Such resistance development occurs with the commonly used tumor necrosis-alpha (TNFα) inhibitors for rheumatoid arthritis, which also have unpleasant side effects, and novel approaches with selective inducers of immune tolerance are under investigation [74].

## 3. Coping with Plasticity

The most effective way to cope with pharmacological plasticity is to control for factors which modify the drug response. This requires the recognition of these factors, as we have sought to emphasize here, to identify biomarkers in order to assess the influence of confounding variables and also to standardize methods and approaches to harmonize between investigators (Table 1).

In recent years, a number of initiatives have been promoted to control variability and poor reproducibility of preclinical research and drug discovery [2]. These encompass target identification, drug screening and animal studies, as well as the more established guidelines for safety testing and clinical trials, such as GLP, GCP and GMP [75,76]. In addition, the use of the adverse outcome pathway (AOP), which takes into account drug-target interactions, signaling processes and reactions of affected organs to try and predict potential adverse effects of drug candidates, is becoming more common [77].

Effective formulation of a new drug candidate can go a long way to improve the delivery, duration of action and reproducibility of the active ingredient. In the past, this required extensive animal experiments, which did not necessarily predict how the formulations would work in humans. Consequently, new techniques are being introduced to be able to predict drug release based on in vitro studies. This topic is discussed in an article by Hajba-Horvath E. et al. [78] in this *IJMS* Special Issue. Using a two-stage dissolution method for in vitro bioavailability and in silico modeling, the authors show that a specialized procedure for preparation of a nanoparticle formulation of the drug valsartan increases bioavailability dramatically, allowing a 60–70% reduction in oral dosing. Approaches to poor drug delivery or rapid metabolism include conjugation with polyethylene glycol or delayed release formulation to extend plasma half-life, and conjugation with targeting molecules, such as antibodies or administration in nanoparticle formulations, as discussed earlier in relation to metabolism.

In an approach to these different issues, we have previously suggested varying dosing regimens to allow for differences in target molecules and disease progression, including specific (targeted), synchronized (for multi-mechanistic drugs), sequential (for drugs with related mechanisms) or simultaneous (for multi-drug) dosing [7]. In keeping with this, a concept has also been proposed recently whereby individual patient-tailored algorithms are introduced to cope with patterns of variability to treatment regimens associated with loss of response to anti-TNFα therapies in rheumatoid arthritis [79]. Additionally, with regard to loss of efficacy due to tachyphylaxis or tolerance, it has been shown that, at least with dopamine D1 agonists for the treatment of Parkinson’s disease and europsychiatric disorders, the introduction of positive allosteric modulators may well overcome this tolerance induction [80].

As alluded to in relation to molecular structure, there is at least one aspect of pharmacological plasticity that can be harnessed for beneficial therapeutic use. Thus, the fact that few therapeutic drugs exert actions on just a single target and interact off-target with other molecules has led to the widespread application of drug repurposing or repositioning for beneficial off-target effects of drugs [81]. This has the advantage to the developing company that costly and time-consuming safety studies are usually not needed because the drugs are already approved for the original clinical indication.

It is crucially important to be able to use robust biomarkers, both for drug responses and plasticity variables at all stages of drug discovery, development and their clinical use. In early research, the biomarkers need to be validated as suitable surrogate markers for clinical responses to permit translation to humans and should adequately assess interactions between the pharmacodynamic responses and confounding variables [82]. For instance, one approach to control for age, diet, disease state, and treatment adherence in drug response, particularly in elderly patients exposed to polypharmacy, is pharmacogenetic testing of patients, which can help distinguish associations between genetic polymorphisms and therapeutic response [83]. Such complex interactions, however, require careful analysis of large amounts of data. For this reason, a recent article has called for greater integration of the 4Ds: drugs, data, diagnostics and devices, to facilitate improved precision medicine [84]. The authors emphasize that greater cooperation between disciplines is needed to cover the wide variety of technologies, skills and knowledge required, and to adapt development approaches, the time for which varies considerably for drugs and medical devices. The examples they provide include how cooperation was successfully introduced into the development of the closed loop system for diagnosis and treatment of diabetes; and how expanded biomarker development and the use of specific biomarkers for subgroups of patients, coupled with wearable medical devices is beginning to revolutionize precision medicine. Additionally, artificial intelligence (AI) and machine learning (ML) are being introduced to enhance the success of both target-based and phenotypic drug discovery approaches [85].

On the basis of the brief discussion in this article, it would seem that complex integrative approaches, involving algorithms established with AI and ML and a variety of different variables and assessment technologies are going to be essential to address the multiple factors and levels of variation which confound the precision and efficacy, for the individual patient, of the drugs with which they are treated. Consequently, we shall undoubtedly see increasing use of digitalized data and analysis, standardization of procedures and biomarkers in order to control for plasticity of drug responses and enhance the use and efficiency of drug therapy [4]. Thus, the reductionistic approach to scientific methods referred to at the start of this article needs reassessing in order to account for the response of the whole organism, even to drugs with a highly specific mechanism of action.

## Figures and Tables

**Table 1 ijms-23-02068-t001:** Confounding variables which lead to plasticity of drug responses.

Level of Plasticity	Confounding Variables	Mitigation
Molecular structure	Drug structure, target structure and folding	Computerized drug design, neural networks, robust screening methods, allosteric modulators, guidelines, drug repurposing
Gene expression	Drug target molecule polymorphism, receptor molecule sensitivity, epigenetic changes	Tissue expression profiling, genotype-phenotype association, pharmacogenetic testing, AI analyses
Drug metabolism	Enzyme polymorphisms, sex differences, product feedback inhibition, reactive metabolites, diet, exercise, drug–drug interactions and polypharmacy	Formulation and targeting, AOPs, genotyping, biomarkers, AI analyses
Cell plasticity	Phenotype changes, ribosome heterogeneity, wind-up, EMT transition, cell senescence, malignant transformation, disease processes	Varied dosing regimen, genotyping, biomarkers, AI analyses
Tissue plasticity	Cardiovascular-metabolic, neuronal-immune, neuroendocrine and chronobiology, viral-epigenetic, sex, experimental conditions and reagents	Varied dosing regimen, AI analyses, pharmacogenetic testing
Environment	Geographic ethnicity, diet, travel	Gene locus screening, AI analyses
Time	Age and aging, disease stage, drug timing, duration and hysteresis, tachyphylaxis and tolerance	Varied dosing regimen, patient-tailored algorithms, allosteric modulators, pharmacogenetic testing

## Data Availability

Citations used in this article were identified using standard web data bases such as PubMed.

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
