# Peer review of "Factors Determining Plasticity of Responses to Drugs"

_ijms, 2022, doi:10.3390/ijms23042068_

Round 1

Reviewer 1 Report

I highly enjoyed reading this commentary. Timely and on a very important topic. Very well written, organized and comprehensive. I have no comments.

Author Response

Thank you for your positive review

Reviewer 2 Report

This review is devoted to a comprehensive analysis of the plasticity of responses to drugs. The authors consider several levels of plasticity. While reading this review, I had a few questions for the authors.

  1. Section 2.2. Gene expression. Here the authors claim “A major source of variation in drug responses is the individual alteration in the expression of relevant therapy genes” (lines 82-83). They then discuss pharmacogenomics, which includes the study of variation that arises from changes in genes regulating drug metabolism or drug target molecule expression. In my opinion, here it is necessary to consider separately the problems associated with the effect of drugs on the expression of target genes, and genes regulating drug metabolism are considered in section 2.3.

It is also unclear why the Level of plasticity (Table 1) considers Drug metabolizing enzymes (e.g. CYPs), receptor molecules, epigenetic changes. Epigenetic changes in section 2.2. are not mentioned at all. Perhaps it would be appropriate to consider drug-drug interaction here, when one drug can affect the metabolism of another one, including through a change in the expression of drug-metabolizing enzyme genes.

  1. Section 2.4. Cell plasticity. The authors analyze the problem of the diversity of cell phenotypes under various pathological conditions. I did not see the conclusion, how does modern pharmacology take this into account?

In general, the review is interesting, it presents an analysis of key problems of pharmacology and can be recommended for publication.

Author Response

1. Section 2.2 has been renamed Expression of drug target genes, to distinguish this from 2.3 and the content modified to restrict the discussion in this section to drug target molecules.

Drug metabolizing enzymes have been removed from the entry Gene expression in Table 1, which already includes drug interactions (now named drug-drug interactions), under Drug metabolism. Drug-drug interactions are now also mentioned specifically in section 2.3.
Epigenetic changes, including the role of DNA methylation and histone acetylation, are already mentioned at the end of section 2.2, where it is indicated that these are discussed in detail by Kringel et al in the same Special Issue of IJMS [20] and therefore, not addressed further in this review. We have expanded this sentence to draw attention to the mechanisms discussed in this other review article.

2. The penultimate paragraph in section 2.4 has been expanded to address this point, as follows:
“….resulting in pronounced alterations in the sensitivity of these cells to drug therapy. As a consequence, responses to drugs can vary with age, with the stage of a disease and with the time of dosing. These aspects of time and disease-related drug susceptibility are discussed below. Essentially, this variability means that doses of the same drug may need to be altered with time or that different drugs will need to be used sequentially at different stages of the disease, as discussed in an earlier review [7]."

Reviewer 3 Report

Line 41 please change facilitatedto to facilitated to

Line 42 ''was made in a recent article [7]'' seems not coherent a thing to mention in relation to the rest of the sentence, please consider rephrasing

Line 268 please change East Asia to East Asian 

Author Response

The errors in lines 41 and 42 arose because of a typing mistake and the missing section has been added.
East Asia has been changed to East Asian 

Reviewer 4 Report

Factors determining plasticity of response to drugs

Parnham and Kricker discussed the issue of drug response plasticity and attribute it to different factors including pharmacology, drug target interactions and biological processes.

Major comments:

  • It is unclear the definition of drug response plasticity and how it is different from drug resistance? In Table 1, several molecular mechanisms are mentioned but I think many of them are related to drug resistance, i.e. some patients are genetically predisposed to certain drugs while some other patients are more resistant.
  • The standardization and harmonization methods for integrating different types of confounding factors are described insufficiently.

Minor comments:

  • typos are found, e.g. line 36: ‘enhance’ -> ‘to enhance’; line 37: ‘science is essentially reductionist’ -> ‘science is reductionism’ or ‘scientists are reductionlists’; line 41: ‘facilitateto’ -> ‘facilitate to’;

Author Response

  1. A definition of the term pharmacological plasticity, as used in this review, is now given in the Introduction on lines 30-31. This covers all mechanisms which affect the outcome of drug administration, including drug metabolism and drug resistance. The latter deals with lack of response to drugs, but plasticity also includes processes which enhance responses, such as greater susceptibility of a specific sex, decreased metabolism due to enzyme phenotypes or sensitisation of a receptor or cell type by a pathological process, such as inflammation.
  2. As written at the end of the Introduction, the aim of this article is to provide “an introduction to the subject, together with approaches taken to limit plasticity. The discussion is by no means comprehensive but offers a bird’s eye view and is intended to encourage further debate.” In view of this aim, we have avoided going into details on individual points, apart from providing illustrative examples. Consequently, we consider that, since we are not experts in AI or data mining, the references cited can best be viewed by the reader to gain further details on approaches. To facilitate this, we have added additional text towards the end of section 3 after the citation of reference [84] as follows: “The authors emphasise that greater cooperation between disciplines is needed to cover the wide variety of technologies, skills and knowledge required and to adapt development approaches, the time for which varies considerably for drugs and medical devices. The examples they provide include how cooperation was successfully introduced into the development of the closed loop system for diagnosis and treatment of diabetes; and how expanded biomarker development and the use of specific biomarkers for subgroups of patients, coupled with wearable medical devices is beginning to revolutionize precision medicine.”

  3. The errors indicated have all been corrected.